# Utility of Contemporary Health Screening in the Diagnosis of Bladder Cancer

**DOI:** 10.3390/diagnostics12051040

**Published:** 2022-04-21

**Authors:** Chung-Un Lee, Wan Song, Michael Jakun Koo, Youngjun Boo, Jae-Hoon Chung, Minyong Kang, Hyun-Hwan Sung, Hwang-Gyun Jeon, Byong-Chang Jeong, Seong-Il Seo, Hyun-Moo Lee, Jeongyun Jeong, SeongSoo Jeon

**Affiliations:** 1Department of Urology, Samsung Medical Center, Sungkyunkwan University School of Medicine, Seoul 06351, Korea; iatronices@naver.com (C.-U.L.); wan.song@samsung.com (W.S.); michaelkoo12@gmail.com (M.J.K.); youngjun91.boo@samsung.com (Y.B.); dr.jhchung@gmail.com (J.-H.C.); dr.minyong.kang@gmail.com (M.K.); hyunhwan.sung@samsung.com (H.-H.S.); hwanggyun.jeon@samsung.com (H.-G.J.); bc2.jung@samsung.com (B.-C.J.); seongil.seo@samsung.com (S.-I.S.); hyunmoo.lee@samsung.com (H.-M.L.); 2Center for Health Promotion, Samsung Medical Center, Sungkyunkwan University School of Medicine, Seoul 06351, Korea; jyuro.jeong@samsung.com

**Keywords:** bladder cancer, health screening, detection, diagnosis

## Abstract

Background: To evaluate the utility of contemporary health screening (HS) in the diagnosis of bladder cancer (BCa). Methods: We retrospectively reviewed 279,683 individuals who underwent HS between February 1995 and April 2015. Among these individuals, 74 were diagnosed with BCa within a year after the HS and were included in the analysis. Screen-detected BCa was defined as when a referral was made to a urologist due to microscopic hematuria (MH) on urinalysis, abnormal imaging, or any urological symptoms observed at the HS. Screen-undetected BCa was defined as when no referral was made to a urologist because of no abnormality observed at the HS, but a visit to a urological outpatient clinic later was followed by a BCa diagnosis. The incidences of screen-detected BCa and BCa in the Korean population were compared. Clinicopathological characteristics were compared between the screen-detected BCa and screen-undetected BCa groups. Results: The detection rate of BCa was 17.2 per 100,000, which exceeded the 2020 estimated national crude incidence rate of 9.3 per 100,000 by approximately 1.7 times. Among the 74 patients diagnosed with BCa within a year after HS, 48 (64.9%) had screen-detected BCa. The screen-detected BCa group had a higher T stage (*p* = 0.009) and grade (*p* = 0.019) than the screen-undetected BCa group. However, the overall survival was not significantly different between the two groups (*p* = 0.677). A positive correlation between the MH grade and the T stage was identified (*p* = 0.001). Conclusion: Although HS is not focused on BCa screening, contemporary HS can contribute to the detection of BCa.

## 1. Introduction

Bladder cancer (BCa) is the 10th most common cancer worldwide, with approximately 573,000 new cases diagnosed in 2020 [1]. Although mortality due to BCa has decreased globally due to early detection, improved surgical outcomes, and a new paradigm of treatment, the incidence of BCa has steadily increased, especially in Europe and other developed countries [2]. Generally, BCa is present as a superficial tumor in 75% of patients. However, 60–80% of patients experience tumor recurrence after transurethral resection of the bladder tumor (TUR-BT) and 15% of patients eventually progress to higher stages and grades [3]. In addition, approximately one-fourth of patients are diagnosed with muscle-invasive BCa, and 20–40% of them have lymph node metastasis at diagnosis [4].

In a meta-analysis to analyze the cause of BCa from 1995 to 2015, 81.8% of BCa cases could be attributed to known preventable causes, and only 7% of BCa cases arose from heritable genetic influence [5]. Smoking and occupational exposure are reported as the first and second greatest preventable risk factors for BCa [6,7]. Therefore, several efforts such as smoking cessation campaigns have been conducted to create awareness and reduce the incidence of BCa; however, these have not been successful, resulting in a significant socioeconomic burden [8,9].

Early diagnosis of BCa is necessary for successful treatment and reduction of the socioeconomic burden caused by BCa. Health screening (HS) is the only method for making an early diagnosis of BCa before the appearance of recognizable symptoms or signs. Currently, screening tests for BCa are known as identifying hematuria with a urine dipstick or microscopic urinalysis, urine cytology, and tests for urine biomarker. The current HS does not include tests for detecting BCa because the diagnostic accuracy of the screening tests for BCa is inadequate [4]. However, HS was not focused on screening for BCa, we observed some patients who were diagnosed with BCa after being transferred from the HS center for further evaluation. Therefore, we hypothesized that although HS is not focused on BCa screening, HS may contribute to diagnosing BCa. In this study, we evaluated the utility of contemporary HS in the diagnosis of BCa.

## 2. Materials and Methods

### 2.1. Ethics Approval

This study was approved by the Institutional Review Board of Samsung Medical Center (IRB No. 2016-09-037), and the requirement for informed consent was waived due to the retrospective nature of this study. All study protocols were performed in accordance with the principles of the Declaration of Helsinki.

### 2.2. Study Population

We retrospectively reviewed the medical records of individuals aged >20 years who underwent HS between February 1995 and April 2015 at our institution. We identified patients who were newly diagnosed with BCa after a TUR-BT within a year after the HS. We limited the period between the HS and the diagnosis of BCa to a year to confirm the impact of the HS. Patients who did not undergo HS in the previous year, who had a history of BCa, and concomitant upper tract urothelial carcinoma were excluded from the analysis.

The included patients were divided into two groups: screen-detected BCa (SDB) and screen-undetected BCa (SUB). SDB was defined as when a referral was made to a urological outpatient clinic for further evaluation due to abnormal findings on urinalysis with microscopy, abnormal findings or suspicious lesions on imaging, and/or urological symptoms such as frequency, dysuria, and urgency, which was finally diagnosed as BCa. Microscopic hematuria (MH) was defined as ≥3 red blood cell (RBC)/high-power field (HPF) on urinalysis. SUB was defined as when no referral was made to a urological outpatient clinic because there was no abnormality or urinary symptoms observed on examination, but a visit to a urological outpatient clinic later was followed by a BCa diagnosis.

### 2.3. HS Protocol

The HS was initiated for those who had applied to the program, and the examinations varied depending on individual demands. For the urological examination, the HS system routinely took the patients’ medical history and performed urinalysis with microscopy. For individuals who wanted an imaging workup, most underwent abdominal ultrasonography (US) and some underwent computed tomography (CT) of the abdomen and pelvis in place of or in addition to abdominal US based on the demand of the patients.

### 2.4. Outpatient Clinic Protocol

For all patients referred to the urological outpatient clinics, urine cytology, CT urography, and cystoscopy were performed to identify bladder tumors. TUR-BT was performed if the bladder tumor was confirmed. Pathologic staging and tumor grading were determined according to the 2010 TNM classification of the American Joint Committee on Cancer (AJCC) [10] and the 2004 World Health Organization (WHO)/International Society of Urologic Pathology consensus classification [11].

### 2.5. Variables Included in the Study

The incidence of SDB and BCa in a Korean population were compared. The patients’ age, sex, body mass index, history, anticoagulant use, follow-up period, and date of death were collected to investigate the characteristics of the SDB and SUB groups. Furthermore, we investigated the relationship between the MH count and the pathologic T stage after the TUR-BT.

### 2.6. Statistical Analysis

Continuous data are presented as the mean (standard deviation, SD), and categorical data are presented as absolute values (percentages). An independent t-test for continuous variables and Pearson’s chi-square test were used to compare categorical data. Kaplan-Meier analysis was used to analyze the overall survival in the SDB and SUB groups. Linear-by-linear association tests were used to analyze correlations between the pathological T stage after the TUR-BT and the MH count. All statistical analyses were performed using IBM SPSS Statistics for Windows (version 23.0; IBM Corp., Armonk, NY, USA). Statistical significance was set at *p* < 0.05.

## 3. Results

### 3.1. Enrollment

Among the 279,683 individuals in the HS program, 150,390 were men and 129,293 were women. Of these, 97 were newly diagnosed with BCa after the HS. Among these patients, the 22 who did not undergo HS in the previous year and one who had concomitant ureter cancer were excluded from the analysis. Therefore, 74 patients were included in the analysis.

### 3.2. Patients’ Characteristics

Among the 74 patients, the proportion of the SDB group was 64.9% (n = 48). Baseline characteristics are summarized in Table 1. Most patients were men, constituting 92.3% of the SUB group and 89.6% of the SDB group. All variables were comparable between the groups, except for the time from the screening to the BCa diagnosis. The time from the screening to the BCa diagnosis was significantly shorter in the SDB group (4.0 ± 3.4 vs. 6.0 ± 3.8, *p* = 0.031). The rate of anticoagulant use was not significantly different between the two groups.

### 3.3. Reasons for Being Referred to a Urologist

Of the 48 SDB patients, 44 (91.7%) were referred due to MH. Six patients (12.5%) had lower urinary tract symptoms (LUTS): five (10.4%) had concomitant MH, and one patient (2.1%) had LUTS only. Image-identified bladder tumors were noted in three patients (6.3%). Of these, two were found by CT scan of the abdomen and pelvis and one was found by abdominal US. Among the 26 SUB patients, the courses that led to a diagnosis of BCa included the presentation of gross hematuria (GH), which was defined as blood visible in the urine reported by the patient or physician, in 13 (50.0%), LUTS in 4 (15.3%), and incidentally detected BCa during a workup for other diseases in 9 (34.6%) who underwent CT scan of the abdomen and pelvis.

### 3.4. Comparison with a Korean Population

In our cohort, the overall detection rate for BCa by HS was 17.2 per 100,000, which exceeded the estimated national crude incidence of 9.3 per 100,000 in 2020 by approximately 1.7 times [12]. This higher incidence of BCa in the HS cohort was also observed in the analyses according to sex. Among the 48 patients with SDB, 89.6% (n = 43) were men and 10.4% (n = 5) were women. The detection rate of BCa was 28.6 per 100,000 in men, which was higher than the estimated national crude incidence, 15.0 per 100,000 men [12]. However, there was little difference in the detection rate of BCa in women in the HS cohort and the estimated national crude incidence (3.9 vs. 3.7 per 100,000) [12].

### 3.5. Pathologic Review

Table 2 shows the pathological findings after the TUR-BT in the two groups. The SDB group had higher T stage (*p* = 0.009) and higher grade (*p* = 0.019). The most common T stage was Ta (80.8%) in the SUB group and T1 (43.8%) in the SDB group. The proportion of grade II was similar in the two groups, but the proportion of grade I was lower and the proportion of grade III was higher in the SDB group. Patients with carcinoma in situ (CIS) were found only in the SDB group. However, in the Kaplan-Meier analysis, the overall survival did not differ significantly between the groups (*p* = 0.677) (Figure 1).

### 3.6. Relationship between the MH Grade and the T Stage

Table 3 shows MH grades according to the T stage of BCa. The MH grade was classified as 0–2 RBC/HPF (no MH), 3–20 RBC/HPF, and >20 RBC/HPF. Among the 72 patients, 31 (41.8%) had no MH. A significant difference was detected in the MH grade according to the T stage. In the linear-by-linear association tests, the MH grade tended to increase with the T stage (*p* = 0.001).

## 4. Discussion

Routine screening for BCa is not supported by the current guidelines because of a lack of evidence regarding the effects of screening [4,13,14,15]. Most organizations have focused on the prevention of BCa, not screening for BCa. The U.S. Preventive Services Task Force (USPSTF) recommendation concluded that there was insufficient evidence to evaluate the effectiveness of BCa screening in asymptomatic adults [4], and the American Academy of Family Physicians (AAFP) supports the USPSTF recommendation [13]. The American Cancer Society (ACS) recommends screening for bladder cancer only to individuals with history of bladder cancer, those with certain birth defects of the bladder, and those exposed to certain chemicals at work [14]. The European Association of Urology recommends counseling patients to stop active and avoid passive smoking, informing workers in potentially hazardous workplaces of the potential carcinogenic effects of a number of recognized substances, including the effect of increased duration of exposure to them and the latency periods, and recommending protective measures for the prevention of muscle-invasive BCa [15]. Collectively, they stated that at the time, no major professional organizations recommended routine screening of the general public for the diagnosis of BCa. They recommend no smoking, limiting exposure to certain chemicals in the workplace, drinking plenty of liquids, and eating lots of fruits and vegetables to prevent BCa.

In this study, we investigated the utility of contemporary HS, which was not focused on the diagnosis of BCa. The incidence of SDB exceeded the estimated national incidence of BCa by approximately 1.7 times. Approximately two-thirds of the patients with BCa were detected using the HS program. Although our study showed that the SDB group showed a higher T stage and grade than the SUB group, there was no difference in the overall survival between the SDB and SUB groups. A positive correlation between the MH grade and T stage was identified. These results indicate that the examinations in the HS program are helpful in detecting BCa, despite the fact that the HS program was not intended for BCa detection. We believe that our findings on the effects of the HS program on the BCa detection rate are valuable.

Gonzalez et al. reported that among the 2118 patients who underwent cystoscopy for asymptomatic MH, 25 patients (1.2%) were diagnosed with BCa: all of these cases were non-muscle-invasive BCa [16]. BCa was an uncommon finding on cystoscopy among patients being evaluated for asymptomatic MH. On the other hand, in our study, most of the SDB patients were referred for MH; therefore, cystoscopy should not be ignored in patients with MH.

In a multi-institutional observational cohort study involving 1384 patients with de novo diagnosed BCa, presentation with GH was associated with a more advanced pathological stage. Ramirez et al. demonstrated that the BCa stage at diagnosis for patients presenting with MH was Ta/CIS in 68.8%, T1 in 19.6%, and ≥T2 in 11.6%. The stages of disease at diagnosis for patients presenting with GH were Ta/CIS in 55.9%, T1 in 19.6%, and ≥T2 in 17.9% [17]. The study suggested that earlier detection of BCa before presentation of gross hematuria could improve the survival rate. In our study, most SDB patients presented with MH and showed higher T stage and tumor grade than SUB patients. The difference between the two studies might be due to the study population: our study population was based on HS. Although 50% of SUB patients have visited the urologic clinic due to GH, those patients did not present with any symptoms and/or MH within a year at the HS, which could help explain the higher incidence of lower stage and lower grade BCa detection than those of SDB patients Furthermore, those who visit urologic clinics without exhibiting any symptoms can be attributed to increased personal health concerns. These individuals have led to prompt detections of early stages of BCa. These individuals have presented these individuals have presented themselves to urologic clinics at the slightest hint of symptoms, and therefore could have led to prompt detections of early stages of BCa. In addition, in their study, the severity of hematuria was simply categorized as MH versus gross hematuria, and the severity of MH was not assessed. We sub-classified MH and found a significant difference in the MH grade according to the T stage and a positive correlation between the MH grade and T stage.

The most common features of BCa are painless gross hematuria and irritative symptoms, which were observed in approximately 90% of patients at the time of diagnosis [18]. In our study, 50% of the SUB patients visited a urology clinic for gross hematuria. The remaining 50% of SUB patients were diagnosed with BCa either via LUTS (15.3%) or a workup for other diseases (34.6%). A relatively low proportion of patients with SUB had gross hematuria as the first presentation, suggesting that some BCa cases could have been diagnosed through HS before gross hematuria appeared.

Although the current recommendations do not support routine BCa screening, some studies have suggested the importance of BCa screening in high-risk patients [19,20]. Krabbe et al. suggested that men older than 60 years with a smoking history of >30 PY had incidence rates of more than 2/1000 person-years, which could serve as an excellent population for screening trials [19]. Also, Sun et al. suggested that BCa screening may be important in high-risk populations. [20] In a systematic review, delay in diagnosis or treatment of BCa is associated with poorer outcomes [21]. Delay in diagnosis or treatment affects survival as an independent variable and is related to higher pathological stage. The above studies support the need for early diagnosis of BCa in high-risk patients, and our study also showed the need for HS by showing that HS is helpful for the early diagnosis of BCa.

In the HS program, urine cytology that was highly recommended for patients suspected of having BCa was not included. For HS, routine checks for urine cytology are not supported by current recommendations [4,15]. However, it may be beneficial to check urine cytology in addition to urinalysis with microscopy in an HS program for a population with well-known BCa risk factors (e.g., smoking, occupational exposure to carcinogens, male sex, older age, white race, or family history of BCa). In a similar context, there are some non-invasive molecular markers present in the blood, serum, plasma, and/or urine for detecting BCa [22]. Madeb et al. suggested second-line testing for screening BCa using a combination of urinary molecular markers, including BCLA-4, NMP-22, BTA-stat, Survivin, FISH, Telomerase, ImmunoCyt, and UroVysion. [23] Although it is difficult to apply these methods instantly, we think that their use in high-risk populations with urine cytology will be of advantage in screening for BCa.

This study has some limitations. First, because of its retrospective nature, selection bias may exist. Second, we could not weigh the carcinogens such as smoking and chemical exposure. It might be though that carcinogens could affect the incidence of BCa in HS population, unfortunately, our study did not separate smokers and chemically exposed patients from the HS cohort; therefore, we could not draw conclusions about the relationship between HS and carcinogens. Third, the 1-year follow-up period was relatively short for evaluating the utility of HS; some patients might have been diagnosed with BCa more than a year after HS. Lastly, we only compared the crude incidence rate between our study and the expected national incidence rate, not the age standardization incidence rate, which might affect the results of our study. Despite these limitations, we believe that the results of this study demonstrate the utility of an ordinary HS for the diagnosis of BCa.

## 5. Conclusions

Although the HS program is not focused on detecting BCa, the incidence of SDB exceeded the estimated Korean crude incidence of BCa by 1.7-fold. We identified a tendency of the MH grade increasing as the T stage increased. These results suggest that HS could contribute to detecting BCa and referral of examinees to urologists when abnormalities are observed in the HS program is imperative.

## Figures and Tables

**Figure 1 diagnostics-12-01040-f001:**
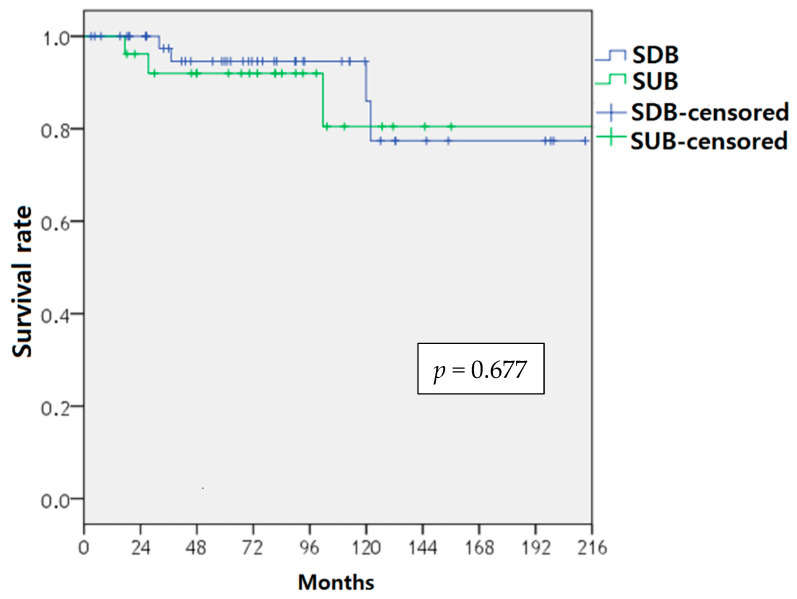
Overall survival of patients with screen-detected bladder cancer and screen-undetected bladder cancer estimated by Kaplan-Meier analysis. SDB, screen-detected bladder cancer; SUB, screen-undetected bladder cancer.

**Table 1 diagnostics-12-01040-t001:** Baseline characteristics of patients with screen-detected bladder cancer and screen-undetected bladder cancer.

	SDB Patients (n = 48)	SUB Patients (n = 26)	*p*
Age, yrs	61.0 ± 8.9	57.0 ± 9.7	0.079
Sex, male, n (%)	43 (89.6)	24 (92.3)	0.702
BMI (kg/m^2^)	24.3 ± 4.6	25.4 ± 3.1	0.265
Follow-up duration (mo)	79.5 ± 54.9	84 ± 51.9	0.732
Time from the screening to the BCa diagnosis (mo)	4.0 ± 3.4	6.0 ± 3.8	0.031
History, n (%)			
Hypertension	15 (31.2)	10 (38.5)	0.531
Diabetes mellitus	4 (8.3)	5 (19.2)	0.171
Tuberculosis	1 (2.1)	2 (7.7)	0.243
Hepatitis	1 (2.1)	0 (0)	0.459
Cerebrovascular accident	1 (2.1)	0 (0)	0.459
Cardiovascular disease	8 (16.7)	7 (26.9)	0.259
Use of anticoagulants, n (%)			0.890
Aspirin	14 (29.1)	9 (34.6)	
Clopidogrel	4 (8.3)	2 (7.7)	

BMI, body mass index; BCa, bladder cancer; SDB, screen-detected bladder cancer; SUB, screen-undetected bladder cancer.

**Table 2 diagnostics-12-01040-t002:** Pathology findings after transurethral resection of bladder tumors of screen-detected bladder cancer and screen-undetected bladder cancer.

	SDB Patients (n = 48)	SUB Patients (n = 26)	*p*
Pathologic stage, n (%)			
T stage			0.009
A	20 (41.7)	21 (80.8)	
I	21 (43.8)	4 (15.4)	
II	5 (10.4)	1 (3.8)	
Grade			0.019
I	6 (12.5)	9 (34.6)	
II	24 (50.0)	15 (57.7)	
III	15 (31.3)	2 (7.7)	
CIS	4 (8.3)	0 (0.0)	0.130

CIS, carcinoma in situ; SDB, screen-detected bladder cancer; SUB, screen-undetected bladder cancer.

**Table 3 diagnostics-12-01040-t003:** Correlation between pathological T stage after transurethral resection of bladder tumor and microscopic hematuria grade.

		Microscopic Hematuria Grade			*p*
		0–2	3–20	≥21	
T stage	A	23	13	5	0.001
	I	7	11	7	
	II	1	1	4	

## Data Availability

The dataset used and/or analyzed during the current study is available from the corresponding author upon reasonable request.

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
