# Peer review of "Utility of Contemporary Health Screening in the Diagnosis of Bladder Cancer"

_diagnostics, 2022, doi:10.3390/diagnostics12051040_

Round 1

Reviewer 1 Report

The manuscript presents a retrospective study for examining the hypothesis that heath screening may improve the diagnosis of bladder cancer. Although the author provided evidence to prove that health screening could improve the detection of bladder cancer, the scale of this research is too small to be published. 

The author should add more content in their study. For instance, creating more hypothesis, describing more details from the data, performing different analysis to provide more perspective.

Author Response

 We totally agree with your opinion. We don't think the scale of our study is large enough. However, we believe that this study was able to derive the effectiveness of health screening for bladder cancer.

 We plan to do further research on  bladder cancer in health screening with larger population.

Reviewer 2 Report

I have no significant objections,
I think it is an interesting and well-written work

Author Response

We thank your constructive and insightful comments.

Reviewer 3 Report

The study by Lee and Song et al shows how general health screening can be used to detect bladder cancer. Overall, the study is clear, well-organized and written. I only have a few suggestions to further improve the article:

  1. Abstract –decrease the number of abbreviations and explain them.
  2. Introduction - further discuss the methods of BCa screening currently used in the clinics and proposals for new screen methods to be implemented.
  3. Results – The SDB group has higher grade and stage in comparison with the SUB group. Please add a possible explanation for this observation.
  4. Results - Please add an analysis comparing subjects exposed to risk factors such as tobacco vs. subjects not exposed to BCa risk factors.

Author Response

Thank you for insightful review.

  1. Abstract –decrease the number of abbreviations and explain them.

 Some abbreviations of this abstract may be unfamiliar to readers who are new to this paper. Therefore, we decrease the number of abbreviations and described as full-term.

  1. Introduction - further discuss the methods of BCa screening currently used in the clinics and proposals for new screen methods to be implemented.

 Description of current guidelines for screening bladder cancer is important. We recognized and described them in the “Introduction” part. Please see revised manuscript page 2 starting with sentence “Currently, screening tests for BCa are known…”.

  1. Results – The SDB group has higher grade and stage in comparison with the SUB group. Please add a possible explanation for this observation.

 There was a study with different results than that of ours, and we have aptly described such study’s results. Also, we explain the possible reasons in “Discussion” part. Please see revised manuscript page 7 starting with sentence “Although 50% of patients with GH visited …”

  1. Results - Please add an analysis comparing subjects exposed to risk factors such as tobacco vs. subjects not exposed to BCa risk factors.

 We absolutely agree with your opinion. Smoking is a well-known risk factor for development of BCa, and we also assume that smoking has affected the development of BCa in our cohort. Unfortunately, we did not analyze about smoking in this study and we described in “Discussion” part. Please see revised manuscript page 7 starting with sentence “It might be thought that carcinogens could affect the incidence of BCa…”. We plan to do further research on smoking and bladder cancer in health screening.

This manuscript is a resubmission of an earlier submission. The following is a list of the peer review reports and author responses from that submission.